# An Exploration of Relationships among Thermal Insulation, Area Factor and Air Gap of Male Chinese Ethnic Costumes

**DOI:** 10.3390/polym12061302

**Published:** 2020-06-06

**Authors:** Ying Ke, Faming Wang

**Affiliations:** 1Jiangsu Non-material Culture Heritage Research Base, Jiangnan University, Wuxi 214122, China; keying@jiangnan.edu.cn; 2School of Architecture and Art, Central South University, Changsha 410083, China

**Keywords:** local thermal insulation, air gap, clothing area factor, ethnic costume, 3D body scanner

## Abstract

The present study investigated total and local thermal insulations of 39 sets of male Chinese ethnic costumes. Total and local clothing area factor, air gap size and air volume were determined by a 3D body scanner. Relationships between thermal insulation and air gap for the whole body, as well as local body parts, were explored. Correlations of both the total and local clothing area factor with the intrinsic insulation were also developed. Results demonstrated that the clothing total thermal insulation first increased with the increasing air gap size/air volume, followed by a decrease when the air gap size/air volume exceeded 37.8 mm/55.8 dm^3^. Similarly, it was also found that parabolic relationships widely existed between the local thermal insulation and local air gap at each body part. Our research findings provide a comprehensive database for predicting both global and local thermal comfort of male Chinese minority groups.

## 1. Introduction

Thermal insulation plays an important role in determining clothing’s thermal comfort and human thermal stress [1,2]. Existing studies have demonstrated that clothing’s thermal insulation is largely affected by wind speed, wind direction, human body movement, body posture and sweating, and such clothing physical parameters as fabric properties (e.g., thickness, weight, density and air permeability), clothing area factor and clothing design features (e.g., covering area, clothing combinations and apertures, wearing style, air gap size and distribution and air volume) [3,4,5,6,7,8,9,10,11,12]. Of all aforementioned factors, the air layer underneath the garment has a much greater effect on clothing insulation than basic fabric properties (e.g., thickness and thermal conductivity) [13,14].

Many documented studies have shown that clothing’s thermal insulation is affected by the fit. Generally, loose-fitting clothing provides higher thermal insulation than tight-fitting clothing [3,15]. The clothing thermal insulation firstly increased with the air gap size (which was estimated by a circumference model), and it started to decrease as the air gap size exceeded a certain value, e.g., 7–8 mm [15,16]. This was mainly due to the occurrence of natural/forced convection within the clothing microclimate. Recent advanced technology in the three dimensional (3D) scanning technique has made possible the study of air layer distribution in clothing and thereby, the relationship between clothing air gap and heat transfer parameters could be explored. The relationship between the air gap distribution and the burn pattern of male and female protective clothing exposed to flash fire was investigated using the 3D body scanning technology [17,18]. Findings demonstrated that skin burn injuries were found at body locations covered with small air gaps. Lu et al. [12,19] quantified the air layer of thermal protective coveralls and analyzed its relationship with the skin burn pattern under hot liquid spray exposures. In general, clothing with bigger air gaps provided better thermal protective performance. More severe skin burns were located at those body parts with smaller air gaps. Lee et al. [14] explored the relationship between the air volume and clothing thermal insulation using the phase-shifting moiré topography. Thermal insulation at the upper body decreased when the microclimate air volume was higher than 7 dm^3^. Li et al. [20] explored the relationship between the air gap and clothing thermal insulation and found that thermal insulation of shirts increased with air gap sizes but began to decrease when the air gap was higher than a certain value, although the cubic regression developed in their work was questionable from a physical viewpoint.

Almost all published studies on clothing heat transfer properties have been focused on Western-style clothing and thermal protective clothing. A limited number of studies have investigated thermal properties of non-Western clothing. Sung [13] determined thermal insulation of Korean costumes and found that the total thermal insulation of Korean costumes was normally greater than that of Western clothing at the lower body part. Al-ajmi et al. [21] measured thermal insulation of the Arabian Gulf traditional costumes and their clothing area factor. The generally accepted empirical formula for predicting clothing area factor based on Western style clothing was invalid for Arabian Gulf costumes. For non-Western clothing, the relation of clothing area factor to clothing insulation was quite different from those developed based on Western clothing [22]. Particularly, costumes from Chinese minority ethnic groups differed greatly in terms of clothing design structure, fitting and wearing styles compared with aforementioned garments. Western clothing is often designed to fit the body perfectly whereas Chinese ethnic minority costumes are loosely fitted. Thus, it is expected that the air gap distribution of Chinese costumes would differ distinctly from that of Western clothing and other non-Western clothing.

Presently, there is a lack of comprehensive study on the determination of thermal properties of Chinese style ethnic costumes. Moreover, the clothing local thermal properties have not been systematically examined yet. The determination of overall and local thermal properties of Chinese ethnic costumes contributes to the body of knowledge about improvement of overall and local thermal comfort of minority groups and thereby, their work performance and health could be enhanced. Hence, in this study, both the total and local air gap, clothing area factor and thermal insulation of 39 sets of Male Chinese ethnic costumes were determined using a 3D body scanner and a multi-segment thermal manikin. The relationship between clothing air gap/clothing area factor and thermal insulation was comprehensively explored. The results of this study further extended the clothing database presented in such standards as ASHRAE55-2013 [23], ISO 7730-2005 [1], and ISO 9920-2009 [24].

## 2. Materials and Methods

### 2.1. Clothing Ensembles

Thirty-nine sets of male ethnic costumes were selected from 39 Chinese ethnic minority groups. They were specially made based on the dimensions of the ‘Newton’ manikin and could be divided into 6 groups, according to their design features. Group 1 (G1) is two-piece garments with short-sleeve shirts or short dresses; group 2 (G2) is robes; group 3 (G3) includes short gowns and trousers or long dresses; group 4 (G4) consists of long-sleeve coats and trousers with decorations at the chest and the legs; group 5 (G5) comprises robes and trousers; group 6 (G6) is multilayer ensembles. The Bonan costume has two wearing ways: Bonan1 (i.e., wearing the left sleeve only) and Bonan2 (i.e., wearing both sleeves). Table 1 shows the physical properties of 39 sets of tested costumes.

### 2.2. Thermal Manikin

A 34-segment ‘Newton’ thermal manikin (Thermetrics LLC, Seattle, WA, USA) was used to determine the total and local thermal insulation. The manikin’s segmental surface temperature or the heating power can be controlled individually. Among all 34 body segments, two segments were designed for seating purposes and thus the heating function of the two segments were disabled in our study. The remaining 32 segments were divided into 11 main body parts [25]. Total and local thermal insulations at those 11 body parts were reported accordingly. The manikin surface temperature and the heat flux generated during the experiment at each body segment were recorded by the ThermDAC^®^ software (Thermetrics LLC, Seattle, WA, USA).

### 2.3. Three-Dimensional Body Scanner

To capture the 3D body shape and clothing profile, a VITUS Smart 3D whole-body laser scanner (Human Solutions GmbH, Kaiserslautern, Germany) was used. This scanner is a non-contact optical measuring system capable of rapidly generating a 360° representation of the surface geometry of an object. The scanned object, formed by point cloud data, can be rotated, resized and sliced. A model mannequin, showing almost the same size of the ‘Newton’ thermal manikin, was used to represent the clothing air layer between the thermal manikin and tested clothing ensembles. The nude mannequin was first scanned, and then the dressed mannequin was scanned with the same position and posture. Each garment was scanned three times, by dressing and undressing. The raw data with a format of “.wrl” were outputted for the determination of the air gap and the clothing area factor.

### 2.4. Determination of the Air Gap and Clothing Area Factor

To determine the clothing air gap, as well as to compute the clothing area factor, the Geomagic Qualify 12.0 software (Geomagic Inc., Morrisville, NC) was used for data analysis. There were some missing areas in the scans where cameras could not capture the data. To accurately determine the air gap between clothing and the manikin body, an integral and smooth body surface was necessary. The scan data were firstly meshed, and then they were rewrapped, and the holes were filled by using the ‘Fill Holes’ tools. Subsequently, the healing wizard function was applied, and the optimized high-quality mesh model (including holes filling, mesh repair, smoothing and scrubbing functions, etc.) was exported (see Figure 1a,b).

To calculate the air gap size, air volume and clothing area factor, both nude and clothed scans were required to be overlapped and aligned as accurately as possible. Accurate alignment required minimal changes in the position of the nude and clothed manikin between scans. The two scans could be aligned by several points of nodules and then slightly shifted the x, y and z coordinates of the clothed scan, to perfectly align the two scans by the alignment algorithm. Body parts were sliced according to the body separation of the thermal manikin. Both total and local air gap distributions over the body surface were presented with different color bars (see Figure 1c). In addition, both total and local air volumes entrapped in clothing were calculated. The surface area of the whole body and local body parts was also reported.

### 2.5. Calculations

Total thermal insulation of the boundary air layer and each costume was calculated by the parallel method, using the following equation [24].
(1)$$I_{t}=\frac{\left(T_{manikin}-T_{a}\right)\times A}{0.155\times \sum_{i=1}^{32}\left(H_{i}\times A_{i}\right)}$$
where *I_t_* is the clothing total thermal insulation, clo; *A_i_* is the surface area of the segment *i*, m^2^; *A* is the total surface area of the manikin, m^2^; *T_manikin_* is the manikin surface temperature (°C); *T_a_* is the air temperature, °C; and *H_i_* is the observed dry heat loss at the segment, *i*, W/m^2^.

Local thermal insulation of each body part was calculated by the global method [24], as shown in Equation (2) [24]
(2)$$I_{t,j}=\frac{\left(\sum \frac{A_{i}\times T_{manikin,i}}{A_{j}}\right)-T_{a}}{0.155\times \sum \frac{A_{i}\times H_{i}}{A_{j}}}$$
where *I_t,j_* is the local thermal insulation of the manikin segment, *i* and the main body part, *j*, respectively, clo; *A_i_* and *A_j_* are the surface areas of the segment, *i*, and the body part, *j*, respectively, m^2^; *T_manikin,i_* and *T_a_* are the manikin surface temperature of the segment, *i*, and the air temperature respectively, °C; and *H_i_* is the observed dry heat loss at the segment *i*, W/m^2^.

Clothing area factor of each ethnic costume was calculated by using the following equation [24].
(3)$$f_{cl}=\frac{A_{cl}}{A_{n}}$$
where *f_cl_* is the clothing area factor, dimensionless; *A_cl_* is the surface area of the clothed mannequin, m^2^; and *A_n_* is the surface area of the nude mannequin, m^2^.

The intrinsic thermal insulation of each costume was calculated by using Equation (4) [24].
(4)$$I_{cl}=I_{t}-\frac{I_{a}}{f_{cl}}$$
where *I_cl_* is the intrinsic thermal insulation, clo; and *I_a_* is the boundary air layer’s thermal insulation, clo.

### 2.6. Experimental Protocol and Test Conditions

All thermal manikin measurements strictly followed the ISO 15831-2004 standard [26]. The constant temperature mode (i.e., *T_manikin_* = 34.0 °C) was used. The relative humidity (RH) and the air velocity inside the climatic chamber were 50 ± 5% and 0.4 ± 0.1 m/s, respectively. For the selected ethnic costumes, the total thermal insulation, local thermal insulation, and the intrinsic thermal insulation were calculated. The data acquisition frequency was 30 s. The detailed procedure of the determination of the air gap and clothing area factor was described in Section 2.4. Both manikin tests and 3D scanning tests for each ethnic costume were performed three times.

## 3. Results

The boundary air layer’s thermal insulation was 0.50 clo, which was determined on the nude thermal manikin at 0.4 ± 0.1 m/s wind. Table 2 presents the air volume, air gap size, clothing area factor, total thermal insulation, and intrinsic thermal insulation of the 39 ethnic costumes. The air volume lies in the range of 23.0~93.1 dm^3^, the average air gap ranges is 15.0~51.4 mm, and the clothing area factor is in the range of 1.11~1.70. The clothing total thermal insulation has a range of 0.81~1.48 clo, and the intrinsic thermal insulation is 0.35~1.18 clo. The Tibetan costume has the largest air layer (the air volume and air gap size) and clothing area factor, whereas the Li costume shows the smallest air layer and clothing area factor. The Bonan2 exhibits the largest total thermal insulation and intrinsic thermal insulation, whereas the Li shows the lowest thermal insulation. In addition, G1 shows the minimal thermal insulation, while G6 exhibits the greatest thermal insulation.

### 3.1. Thermal Insulation, Clothing Area Factor and Average Air Gap

The relationship between the clothing total thermal insulation and the air gap (i.e., average air gap size and air volume) is presented in Figure 2. It is evident that the clothing total thermal insulation first increases with the increasing of air gap size (air volume), and it begins to decrease when the air volume (or the mean air gap size) reaches 55.8 dm^3^ (or 37.8 mm). To ensure that the total thermal insulation equals that of the boundary air layer (i.e., 0.50 clo) when no clothing layer presents, the regression equation was forced to go through the point (0, 0.50).

The coefficient factor R^2^ is higher than 0.52 and the equations read as follows:(5)$$I_{t}=-0.00016\times V_{cl}^{2}+0.0222\times V_{cl}+0.50$$
(6)$$I_{t}=-0.00047\times d_{air}^{2}+0.0376\times d_{air}+0.50$$
where *I_t_* is the clothing total thermal insulation (clo); *V_cl_* is the clothing air volume (dm^3^); and *d_air_* is the average air gap size (mm).

Figure 3 shows the relationship between clothing intrinsic thermal insulation and clothing area factor. In order to ensure that the clothing area factor equals the unity if there is no clothing dressed, the regression equation was forced to pass the point (0, 1.0). The coefficient factor R^2^ is 0.53, and the equation reads as follows:(7)$$f_{cl}=1.00+0.42\times I_{cl}$$
where *I_cl_* is the clothing intrinsic thermal insulation (clo); and *f_cl_* is the clothing area factor, dimensionless.

Generally, the higher the clothing area factor, the larger the clothing insulation. Of all 39 ethnic costumes, the Tibetan costume had the largest *f_cl_* but a relatively small *I_cl_*. If it was excluded from the database (because Tibetan costume violated the generally accepted relation), then Equation (7) can be expressed as follows (R^2^ = 0.58).
(8)$$f_{cl}=1.00+0.40\times I_{cl}$$

As is shown in Figure 2, the clothing thermal insulation might decrease if the air gap is larger than a certain value, e.g., the thermal insulation of the Tibetan costume is low although its air layer is the largest among all studied costumes. The parabolic equation was developed to characterize the relationship between *I_cl_* and *f_cl_* (see Figure 4), and it reads as follows:(9)$$I_{cl}=-6.543\times f_{cl}^{2}+9.57\times f_{cl}-3.024$$

### 3.2. Clothing Local Thermal Insulation and Local Air Gap

In this study, the head, hands and feet were not covered by clothing. Thus, only eight body parts (i.e., arms, chest, shoulders, abdomen, back, pelvis, thighs and lower legs) were investigated. Figure 5 shows the relationship between the local thermal insulation and local air gap for eight body parts. In general, the local thermal insulation increases with the increasing local air gap size, while it turns to decrease as the local air gap size exceeds a certain level (e.g., the turning point for the thigh and the lower leg is 35.7 and 40.8 mm, respectively). The results are presented in Table 3. In order to ensure that the *I_t,j_* equals the local boundary air layer’s thermal insulation if no clothing is presented, the regression equation was forced to pass the point (0, *I_a,j_*) for all body parts. The prediction quality is judged by the correlation factor R^2^. The equation reads as follows:(10)$$I_{t,j}=a\times d_{air,j}^{2}+b\times d_{air,j}+I_{a,j}$$
where *I_t,j_* is the local clothing thermal insulation (clo); *I_a,j_* is the local insulation of the boundary air layer (clo); and *d_air,j_* is the local air gap size (mm).

### 3.3. Local Intrinsic Thermal Insulation and Local Clothing Area Factor

Figure 6 presents the relationships between the local intrinsic thermal insulation and local clothing area factor for eight body parts. Eight linear regression equations were developed and the results are shown in Table 4. The regression equation was also forced to go through the point (0, 1). The equation reads as follows:(11)$$f_{cl,j}=1.00+a\times I_{cl,j}$$
where *I_cl,i_* is the local clothing intrinsic thermal insulation at the body part, *j*, clo; and *f_cl,j_* is the local clothing area factor at the body part, *j*.

## 4. Discussion

In the present study, both total and local clothing heat transfer properties (i.e., thermal insulation, air gap, and clothing area factor) of 39 ethnic costumes were investigated. Due to the garment construction varieties, the total thermal insulation varied from 0.81 to 1.48 clo, and the intrinsic thermal insulation was found in the range of 0.35~1.18 clo. G1 shows the minimal thermal insulation, while G6 exhibits the maximum thermal insulation. This is because costumes in G1 had the smallest body covering area while G6 had multiple clothing layers. The Bonan costume had two wearing styles. The costume in different wearing styles showed different thermal insulation values, e.g., a lower thermal insulation was presented if only one sleeve of the Bonan costume was worn. This finding is in line with Li et al.’s study [5], in which the thermal insulation of the Tibetan robe ensemble was determined in three different wearing styles. In our study, the air volume and the average air gap size were in the range of 23.0~93.1 dm^3^ and 15.0~51.4 mm, respectively, and the clothing area factor was 1.11~1.70. The Tibetan costume showed the largest air layer and clothing area factor, whereas the Li costume presented the smallest air layer and clothing area factor. According to the clothing database of ISO 9920-2009 [24] and ASHARE 55-2013 [23], it was found that the thermal insulation and clothing area factor of male Chinese ethnic costumes were in between those of the Western clothing and the non-Western clothing.

The effect of garment fit on total thermal insulation has been explored. It was found that the total thermal insulation increased with the increasing air gap size (or air volume), and the increase rate gradually decreased. With the further increasing of the air gap size or the air volume, the total thermal insulation started to decrease. This is consistent with the findings of previous studies [3,14,15,27], i.e., when the air gap is small, the increase in the air gap size or the air volume raises the total thermal insulation because of more entrapped still air. If the air gap reached a certain limit, natural convection and air circulation in the microclimatic air layers occurred. The rate of ventilation depends on the clothing fit, with more ventilation occurring in loose-fitting clothing than in tight-fitting clothing [28]. Therefore, the increment of the thermal insulation with the air gap gradually decreased and a further increasing of the air gap could cause a drop in the total thermal insulation. In addition, the turning point of the air volume and the air gap size to induce a dropping in the total thermal insulation was 55.8 dm^3^ and 37.8 mm, respectively. These two reported values were different from the previous studies. Lee et al. [14] found that clothing’s thermal insulation began to decrease when the air volume was larger than 7 dm^3^. Thermal insulation turned to decrease as the air gap size was larger than 10 mm [15]. However, it should be noted that Chen et al. [15] investigated the effect of garment fit on the upper body’s total thermal insulation. There was no clothing or the same clothing dressed at the lower body. In Chen et al.’s study [15], the air gap and thermal insulation at the upper body were estimated using a circumference model. This may introduce a big error to quantitatively characterize the relationship between the air gap size and clothing thermal insulation. The 3D scanning technique has been recognized as the most reproducible and accurate method for measuring the microclimate air volume among the widely used approaches [29]. Although there are many studies on the air gap determination of the full-body clothing ensemble using 3D body scanning [18,19,30], there is no solid air gap data reported for robes or the dress style ensembles. In this study, we investigated the thermal insulation and quantitatively determined the air gap distribution of various costumes using the 3D body scanning technology. Thus, the results obtained in this study are systematic and persuasive.

Unlike the air layer averaged along the body circumference obtained by the circumference model or the total air volume determined by the vacuum suit method, the application of 3D body scanning could accurately show the air gap distribution and figure out the occurrence locations of convection within the clothing system. The relationship between the local air gap and the local thermal insulation at different body parts was developed. All eight body parts showed significant parabolic correlations but coefficients were different. The local thermal insulation increased with the increasing air gap size, and it began to drop as the air gap size exceeded a certain value. This is similar to the relationship found between the total thermal insulation and the average air gap size. The turning point for the eight body parts was in the range of 7.0 (shoulders) ~43.9 mm (arms). The air gap size at the chest and shoulders was lower than that at the arms, thighs and lower legs, which was in good agreement with Lu et al.’s study [19]. It seems that the turning point for each body part was linearly related to the corresponding average air gap (y=0.9996x+12.103, R^2^ = 0.66). In general, body regions with larger air gap sizes had a larger turning point value. For such body parts as the lower body, loose fitting garments permit a larger extent of air convection and ventilation, and thus the local thermal insulation at those regions was small. The relationship between the thermal insulation and the air gap at different body parts is vital to develop models for predicting human local thermal comfort while wearing ethnic costumes.

With regard to the relationship between local thermal insulation and local air gap size as shown in Table 3, the upper body regions (i.e., arms, chest, shoulder, abdomen and back) showed a higher R^2^ than that of the lower body (i.e., the pelvis, thigh and lower legs). One possible reason is that the pelvis, thighs or lower legs were always covered by multilayer clothing (e.g., garments in G5 and G6). This will add the insulation but may not greatly change the air gap size, and thus decrease the correlation factor (i.e., R^2^ values) of the regression. Considering the long robe/dress style of tested costumes, there might be systematic errors in the calculation of air gaps at thighs and lower legs. In addition, some tested costumes consisted of waist belts, which reduced air gaps at the abdomen and the back. Nevertheless, this may not have reduced the local thermal insulation due to the adding belt, and thus affected the changing trend at those body regions. To clarify this influencing factor, the regression analysis of costumes with and with no belt was performed and the results are shown in Figure 7 and Table 5. It is evident that this improved the correlation factor R^2^ (e.g., 0.60 and 0.73 for the abdomen and the back when the belt was worn). As there was no pronounced change in the air gap at the back among all costumes, only a linear relationship was developed. In the specified air gap intervals, thermal insulation increased with the air gap.

The correlation of clothing area factor *f_cl_* with the intrinsic thermal insulation *I_cl_* was also developed. The equation reads $f_{cl}=1.00+0.42\times I_{cl}$ (0.3 clo < *I_cl_* < 1.2 clo). McCullough developed a similar equation based on Western style clothing and this equation was later adopted by ISO 9920-2009 [24]. The equation reads $f_{cl}=1.00+0.31\times I_{cl}$ (0.3 clo < *I_cl_* < 1.7 clo). It is obvious that our equation is different from McCullough’s equation. This is in line with studies of Al-ajmi et al. [21] and Havenith et al. [22], who found the coefficient was ranged from 0.447 to 0.460. It should be noted that the intrinsic thermal insulation was calculated based on a cylindrical clothing model. The model assumed that clothing was worn as cylinders around different human body parts (which were also described as cylinders with different diameters). This seems true for typical Western style clothing. Non-Western costumes such as Arabian Gulf costumes and Chinese ethnic costumes chosen for this study had a loose fit and long robes/dress styles, so it was expected that there was air movement underneath such loose fitting costumes. Hence, the still boundary air layer may only be around the legs rather than spread all over the outer costumes. Therefore, a large error may exist when calculating the clothing intrinsic thermal insulation. With regard to practical application, the determination of intrinsic thermal insulation should consider the garment construction as well as fabric properties such as drapability.

## 5. Conclusions

The present study investigated both the total and local dry heat transfer properties of 39 sets of male Chinese ethnic costumes. Results demonstrated that the total thermal insulation first increased with the air gap size or air volume, and with a further increasing of the air gap size or the air volume, the thermal insulation turned to decrease. The turning point of the air gap size and the air volume was 37.8 mm and 55.8 dm^3^ respectively. The correlation of local thermal insulation with the local air gap showed similar changing trends. Different body parts exhibited similar regression equations but the coefficients were different. A linear relationship between the clothing area factor and the intrinsic thermal insulation was also developed. It was found that the equation developed for Male Chinese ethnic costumes was different from the one established based on Western clothing. The results provide an extensive database of heat transfer properties of non-Western style garments, which is expected to be a valuable addition to the ASHRAE Standard 55-2013, ISO Standard 7730-2005, and ISO Standard 9920-2009. The research findings will also be helpful to develop heat transfer models for predicting local thermal comfort of Chinese minority groups while wearing ethnic costumes.

## Figures and Tables

**Figure 1 polymers-12-01302-f001:**
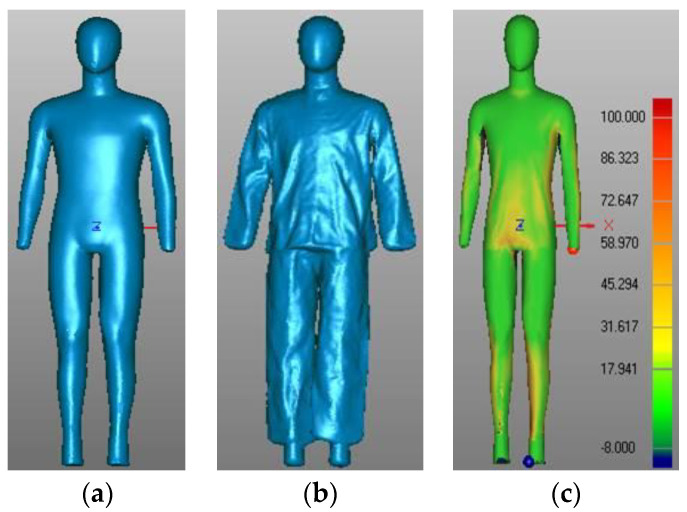
Determination of the air gap distribution: (**a**) scanned nude mannequin; (**b**) scanned clothed mannequin, and (**c**) air gap distribution.

**Figure 2 polymers-12-01302-f002:**
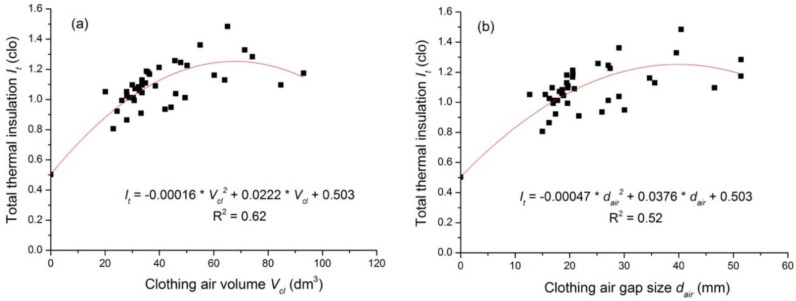
Relationship between the clothing total thermal insulation and (**a**) the air volume; (**b**) the air gap size.

**Figure 3 polymers-12-01302-f003:**
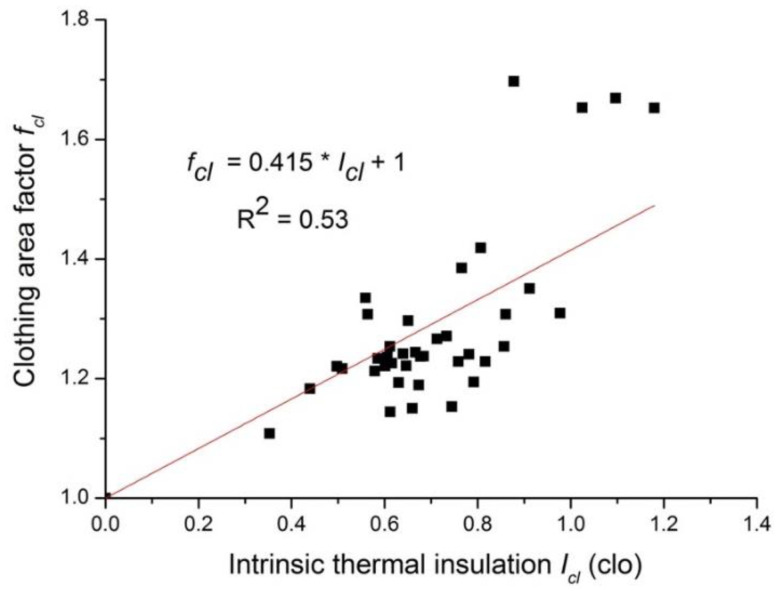
Relationship between the clothing intrinsic thermal insulation and the clothing area factor.

**Figure 4 polymers-12-01302-f004:**
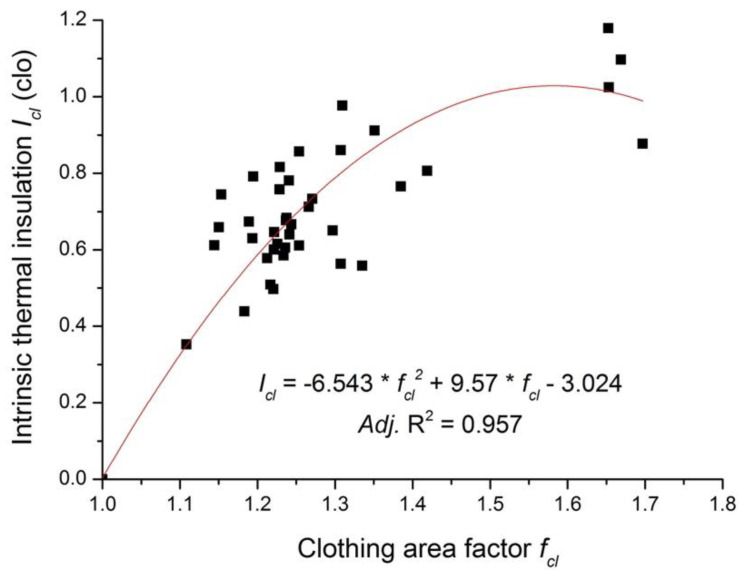
Parabolic correlation of the clothing area factor with the intrinsic thermal insulation.

**Figure 5 polymers-12-01302-f005:**
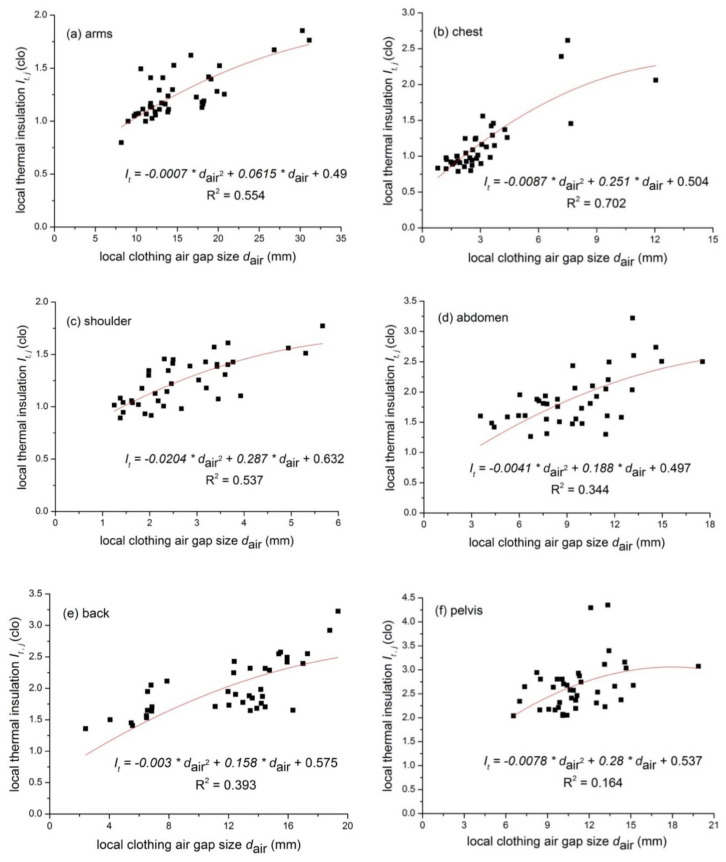
Relationships between the local *I_cl_* and the local *d_air_* for different body parts. (**a**) arms; (**b**) chest; (**c**) shoulders; (**d**) abdomen; (**e**) back; (**f**) pelvis; (**g**) thighs; and (**h**) lower legs.

**Figure 6 polymers-12-01302-f006:**
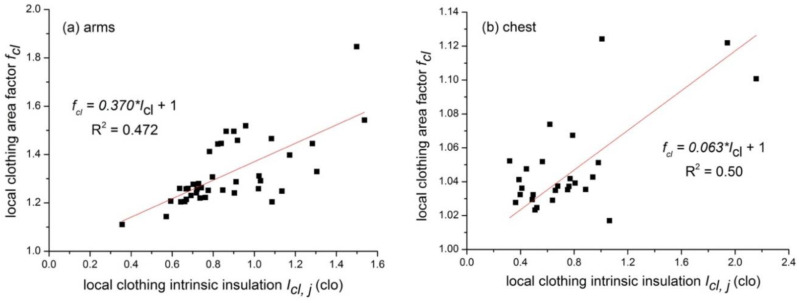
Relationships between the local *I_cl_* and local *f_cl_* for different body parts. (**a**) arms; (**b**) chest; (**c**) shoulders; (**d**) abdomen; (**e**) back; (**f**) pelvis; (**g**) thighs; and (**h**) lower legs.

**Figure 7 polymers-12-01302-f007:**
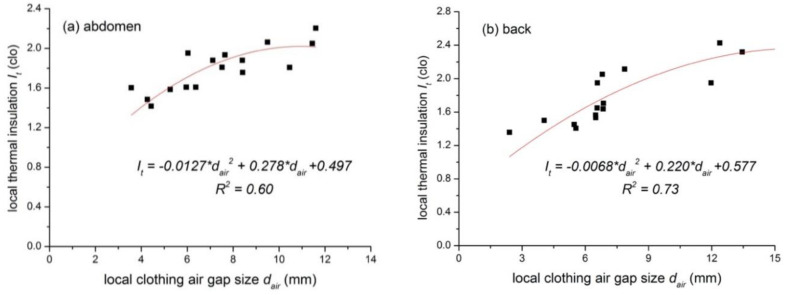
Relationship between the air gap size and the thermal insulation at the abdomen (**a**) and the back (**b**) with the waist belt.

**Table 1 polymers-12-01302-t001:** Characteristics of the 39 sets of male minority ethnic costumes.

Groups	Costume	Components	Materials	Fabric Thickness (mm)	Weight (g)
G1	Li	Short gown	100% polyester	0.58	512.2
		Short dress	100% polyester	0.51	
	Russian	Short sleeve shirt	100% rayon	0.25	477.6
		Trousers	100% polyester	0.58	
	Dai	Short sleeve shirt, trousers	100% rayon	0.36	499.8
G2	Hezhe	Robe	100% polyester	1.62	395.8
	Nu	Robe	55% flax, 45% cotton	0.64	673.6
	Oroqen	Robe	100% polyester	0.81	743.6
	Mongolian	Robe	100% polyester	0.61	773.6
	Tibetan	Robe	100% polyester	0.38	1297.8
G3	Zhuang	Short gown, trousers	100% polyester	0.29	486.2
	Tajik	Short gown, trousers	100% polyester	0.60	992.0
	Pumi	Short gown	100% polyester	0.86	812.0
		Trousers	100% polyester	0.55	
	Va	Short gown	100% polyester	0.64	657.4
		Long dress	78% polyester, 18% rayon, 4% spandex	0.73	
	Jino	Short gown, trousers	100% polyester	0.56	670.4
	Sui	Short gown, trousers	100% polyester	0.70	761.4
	Gelao	Short gown	65% polyester, 35% flax	0.26	470.6
		Trousers	100% polyester	0.44	
	She	Short gown, trousers	96% polyester, 4% spandex	0.82	786.0
	Bouyei	Short gown	65% polyester, 35% flax	0.72	1180.0
		Long dress (up)	100% polyester	0.50	
		Long dress (down)	100% polyester	0.31	
G4	Bai	Short gown, trousers	96% polyester, 4% spandex	0.26	545.0
		Waistcoat	100% polyester	1.73	
	Dongxiang	Short shirt, trousers	100% polyester	0.59	845.0
		Waistcoat	98% polyester, 2% spandex	0.63	
	Yao	Long shirt, apron, trousers	100% polyester	0.64	997.8
	Dong	Short gown	96% polyester, 4% spandex	0.28	928.8
		Waistcoat, trousers	98% polyester, 2% spandex	0.63	
	Nakhi	Trousers	65% polyester, 31% rayon, 4% spandex	0.50	806.4
		Short gown	100% polyester	0.61	
		Waistcoat	100% polyester	0.58	
	Hani	Short gown	54% polyester, 33%cotton, 13% polyamide	0.31	994.0
		Waistcoat, apron, trousers	98% polyester, 2% spandex	0.73	
	Hui	Long shirt, trousers	100% polyester	0.62	975.6
		Waistcoat	100% polyester	0.66	
	Uyghur	Short gown, trousers	96% polyester, 4% spandex	0.30	732.4
		Long waistcoat	10% polyester, 85% cotton, 5% flax	0.70	
	Miao	Robe	50% polyester, 50% polyamide	0.43	934.6
		Waistcoat, trousers	98% polyester, 2% spandex	0.63	
G5	Daur	Robe	96% polyester, 4% spandex	0.85	1066.0
		Trousers	96% polyester, 4% spandex	0.60	
	Mulao	Robe	100% rayon	0.48	669.8
		Trousers	100% polyester	0.30	
	Xibe	Robe, trousers	96% polyester, 4% spandex	0.81	1213.6
	Manchu	Robe	100% rayon	0.47	887.8
		Trousers	100% polyester	0.44	
	Uzbek	Robe	10% polyester, 85% cotton, 5% flax	1.19	1120.4
		Short shirt	100% rayon	0.22	
		Trousers	98% polyester, 2% spandex	0.57	
	Kazakh	Short shirt	100% polyester	0.60	1153.6
		Robe, trousers	100% polyester	0.63	
	Derung	Robe, trousers	100% polyester	0.55	1262.0
		Short shirt	100% polyester	0.60	
	Tujia	Short shirt	55% flax, 45% cotton	0.60	924.4
		Short gown, trousers	98% cotton, 2% spandex	0.52	
G6	Tu	Trousers	100% polyester	0.62	1506
		Short shirt, short gown, waistcoat	98% cotton, 2% spandex	0.63	
	Yi	Short gown, dress smock	100% polyester	0.537	903.0
		Trousers	96% polyester, 4% spandex	0.133	
	Bonan	Long shirt, robe, trousers	100% polyester	0.59	1748.6
	Qiang	Robe	96% polyester, 4% spandex	0.84	1788.8
		Fur waistcoat	96% polyester, 4% spandex	5.09	
		Trousers	62% rayon, 38% polyester	0.65	

**Table 2 polymers-12-01302-t002:** Air volume, air gap size, clothing area factor, total thermal insulation, and intrinsic thermal insulation of the selected ethnical costumes.

Groups	Costume	V_cl_ (dm^3^)	*d_air_* (mm)	*f_cl_*	*I_t_* (clo)	*I_cl_* (clo)
G1	Li	22.99	15.00	1.11	0.81	0.35
	Russian	27.98	16.23	1.18	0.86	0.44
	Dai	24.41	17.42	1.22	0.92	0.51
G2	Hezhe	33.21	21.67	1.22	0.91	0.50
	Nu	42.07	25.92	1.34	0.94	0.56
	Oroqen	44.30	30.05	1.31	0.95	0.56
	Mongolian	46.00	29.05	1.30	1.04	0.65
	Tibetan	93.06	51.39	1.70	1.17	0.88
G3	Zhuang	26.19	16.99	1.23	0.99	0.59
	Tajik	30.76	19.60	1.21	0.99	0.58
	Pumi	30.30	17.19	1.25	1.01	0.61
	Va	49.43	27.08	1.22	1.01	0.60
	Jino	28.06	16.27	1.23	1.03	0.62
	Sui	33.61	18.90	1.24	1.05	0.64
	Gelao	28.93	17.79	1.24	1.01	0.61
	She	20.07	12.67	1.14	1.05	0.61
	Bouyei	84.67	46.54	1.67	1.10	0.80
G4	Bai	27.95	15.53	1.19	1.05	0.63
	Dongxiang	31.07	18.15	1.24	1.07	0.67
	Yao	32.29	18.62	1.24	1.08	0.68
	Dong	33.60	19.64	1.15	1.10	0.66
	Nakhi	34.83	19.70	1.27	1.11	0.71
	Hani	36.37	20.53	1.23	1.17	0.76
	Hui	38.52	20.84	1.24	1.09	0.68
	Uyghur	30.02	16.79	1.19	1.10	0.67
	Miao	50.18	27.41	1.23	1.23	0.82
G5	Daur	32.70	18.61	1.22	1.06	0.65
	Mulao	33.57	19.45	1.27	1.13	0.73
	Xibe	64.02	35.60	1.38	1.13	0.77
	Manchu	60.16	34.63	1.42	1.16	0.81
	Uzbek	35.85	19.47	1.15	1.18	0.74
	Kazazh	39.89	20.56	1.19	1.21	0.79
	Drung	47.83	27.08	1.31	1.25	0.86
	Tujia	35.19	20.58	1.24	1.19	0.78
G6	Tu	45.72	25.20	1.25	1.26	0.86
	Yi	74.20	51.40	1.35	1.28	0.91
	Bonan1	71.39	39.55	1.65	1.33	1.02
	Bonan2	65.07	40.43	1.65	1.48	1.18
	Qiang	54.99	29.03	1.31	1.36	0.98

**Table 3 polymers-12-01302-t003:** Coefficients of the regression equations for eight different body parts.

Body Part	a	b	$I_{a,i}$	R^2^	Turning Point
Arms	−0.0007	0.0615	0.490	0.55	43.9
Chest	−0.0087	0.251	0.504	0.70	14.4
Shoulder	−0.0204	0.287	0.632	0.54	7.0
Abdomen	−0.0041	0.188	0.497	0.34	22.9
Back	−0.003	0.158	0.575	0.39	26.3
Pelvis	−0.008	0.28	0.537	0.16	17.9
Thigh	−0.0014	0.1	0.452	0.30	35.7
Lower leg	−0.0004	0.0326	0.480	0.21	40.8

**Table 4 polymers-12-01302-t004:** Linear regression equations of local *f_cl_* and *I_cl_* for different body parts.

Body Part	a	R^2^
Arms	0.370	0.47
Chest	0.063	0.50
Shoulder	0.044	0.18
Abdomen	0.124	0.44
Back	0.161	0.39
Pelvis	0.048	0.20
Thigh	0.119	0.04
Lower leg	0.84	0.24

**Table 5 polymers-12-01302-t005:** Coefficients of regression equations for the abdomen and back.

Body Part		a	b	$I_{a,i}$	R^2^
Abdomen	No belt	−0.0009	0.139	0.497	0.45
	With belt	−0.0127	0.278	0.497	0.60
Back	No belt	0	0.104	0.575	0.44
	With belt	−0.0068	0.2203	0.575	0.73

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
