# Peer review of "An Exploration of Relationships among Thermal Insulation, Area Factor and Air Gap of Male Chinese Ethnic Costumes"

_polymers, 2020, doi:10.3390/polym12061302_

Round 1

Reviewer 1 Report

2.4: it is unclear how the data was "rewrapped and the holes were filled". Please provide more details, especially about the precision of this method and its validity. How was the mesh model "optimized"? How was the alignement achieved to be "as accurate as possible"?

3 How did the authors determine that the boundary air layer is 0.50 clo?

Table 2: what is the variance in the measurements?

Figure 2 (and following figures): how can the insulation decrease be explained?
If the quadratic model was correct, the thermal insulation would be 0 for an air gap size of around 100 mm - this is not possible.

line 187: What would be the reason to exclude the Tibetan costume from the database?

Author Response

2.4: it is unclear how the data was "rewrapped and the holes were filled". Please provide more details, especially about the precision of this method and its validity. How was the mesh model "optimized"? How was the alignement achieved to be "as accurate as possible"?

REPLY: the holes were filled using the Fill Holes tools. Holes are relatively small and they account for a very small portion of the total surface area. The aim of 'Fill Holes' is to ensure the entire surface is smooth and continuous so that the total surface area could be computed by the software. This method has been widely used in literatures. The optimized mesh model included hole filling, mesh repairing, smoothing, scrubbing etc. to ensure the mesh model was smooth and the mesh quality is good. The alignment was performed using the alignment algorithm in Geomagic Qualify software. Revisions were made in revised manuscript for clearer presentation.

3 How did the authors determine that the boundary air layer is 0.50 clo?

REPLY: This was measured using the manikin when it was nude. We added the description in the revise text.

Table 2: what is the variance in the measurements?

REPLY: the variances are small, for example, manikin tests require the determined clothing insulation should have a SD of <5%. The Geomagic Qualify software gives accurate calculations on the surface area and air volume.

Figure 2 (and following figures): how can the insulation decrease be explained?
If the quadratic model was correct, the thermal insulation would be 0 for an air gap size of around 100 mm - this is not possible.

REPLY: Thanks for your concerns. The model is only valid in the range of air volume indicated in the figures. The minimum insulation is 0.5 clo, which was determined by the manikin, i.e., the boundary air layer's insulation. Insulation could not be 0. Extrapolation of the data should not be performed because most clothing has an air gap of less than 60 mm.

line 187: What would be the reason to exclude the Tibetan costume from the database?

REPLY:Generally the higher the clothing area factor, the larger the clothing insulation. The Tibetan costume showed slightly different trend with other clothing due to its special design (the belt). The belt could significantly reduce the air volume (thereby the insulation), but not the surface area. Hence, it was both included and excluded to show the linear relation between clothing intrinsic insulation and area factor.

Reviewer 2 Report

The presented article describes the relationships between clothing thermal insulation, area factor and air gap of Chinese ethnic minority costumes. Authors focused on method design, calculation and data description. Whole manuscript was sum up in discussion part. It could be see that the authors have worked hard – they tested many costumes and did a lot of calculation part. Unfortunately I do not see any novelty of this work. In introduction authors wrote about another researchers results which they drew the same conclusions. Only difference is in the costume type. Additionally many editorial errors can be found.

However, I think the article can be improved. For this reason I recommend major revision of this article.

Below you can find detailed comments:

  1. Introduction must be improved to show the novelty. Describe for whom the results of these studies are important, maybe they have an impact on the lives of the surveyed minorities. To many times “Chinese ethnic minority” is used. This is the most important for me because I don’t see any novelty of the article but I give the authors chances to fix it.
  2. Why are R2 values so small? Maybe you need to use a different model? Please change it or explain.
  3. In whole article lack of the spacing between parts of the text is incorrectly selected or missing at all. Spaces between subsections, before and after equations, etc. have to be make.
  4. In whole manuscript the description of the drawings should be located above those drawings and not below.
  5. Lines 187-189 => where in this graph and where equation (8) was used? Maybe the equations 7 and 8 were mistaken?
  6. No reference to equations in the text.

Author Response

The presented article describes the relationships between clothing thermal insulation, area factor and air gap of Chinese ethnic minority costumes. Authors focused on method design, calculation and data description. Whole manuscript was sum up in discussion part. It could be see that the authors have worked hard – they tested many costumes and did a lot of calculation part. Unfortunately I do not see any novelty of this work. In introduction authors wrote about another researchers results which they drew the same conclusions. Only difference is in the costume type. Additionally many editorial errors can be found.

However, I think the article can be improved. For this reason I recommend major revision of this article.

REPLY: Thank you for your comments. It's true that the work load reported in the manuscript is huge and we generally spent over a year to finish the work and data analysis. We are sorry that the novelty of our work is not clearly mentioned in our manuscript. In the revised manuscript, we have pointed out the novelty and hopefully revisions have been made satisfactory.

Below you can find detailed comments:

1) Introduction must be improved to show the novelty. Describe for whom the results of these studies are important, maybe they have an impact on the lives of the surveyed minorities. To many times “Chinese ethnic minority” is used. This is the most important for me because I don’t see any novelty of the article but I give the authors chances to fix it.

REPLY: Thanks. We have revised the Introduction and pointed out the novelty. The work contributed to the body of knowledge about improvement of overall and local body thermal comfort of minority groups. Thereby the work performance and healthy of minority groups could also be enhanced.

2) Why are R2 values so small? Maybe you need to use a different model? Please change it or explain.

REPLY: Thanks. It's true that some body parts showed small R2 values. This has been explained in Discussion section: "thighs or lower legs were always covered by multi-layer clothing. This will add the insulation but may not greatly change the air gap size, and thus decrease the correlation factor of the regression. Considering the long robe/dress style of the tested costumes, there might be errors in the calculation of the air gap at thighs and lower legs. In addition, some tested costumes consisted of waist belts, which reduced the air gap at the abdomen and the back. Nevertheless, this may not reduce the total thermal insulation due to the adding belt, and thus affected the changing trend at those body regions."

Based on the generally accepted trend between the air gap size and insulation (insulation increases with air gap size and then started to decrease), other models violated the theory. Therefore, we used the parabolic relation in the manuscript.

3) In whole article lack of the spacing between parts of the text is incorrectly selected or missing at all. Spaces between subsections, before and after equations, etc. have to be make.

REPLY: Thanks, changes were made.

4) In whole manuscript the description of the drawings should be located above those drawings and not below.

REPLY: Revised, thanks.

5) Lines 187-189 => where in this graph and where equation (8) was used? Maybe the equations 7 and 8 were mistaken?

REPLY: Equation 7 was obtained from Figure 3 and Equation 8 was derived from data excluding the Tibetan costume.

6)No reference to equations in the text.

REPLY: Added, thanks. Equations 1-4 were taken from references and the remaining equations were derived from the data presented in this work.

Round 2

Reviewer 1 Report

Thank you for considering the comments

Reviewer 2 Report

The authors have made the corrections I suggested. I am satisfied with the changes. Therefore, I recommend accepting the article for printing.